# Global projections of heat exposure of older adults

Giacomo Falchetta [1,2,3] ✉, Enrica De Cian [1,2,4], Ian Sue Wing [5] & Deborah Carr [6]

The global population is aging at the same time as heat exposures are increasing due to climate change. Age structure, and its biological and socio-economic drivers, determine populations' vulnerability to high temperatures. Here we combine age-stratified demographic projections with downscaled temperature projections to mid-century and find that chronic exposure to heat doubles across all warming scenarios. Moreover, >23% of the global population aged 69+ will inhabit climates whose 95th percentile of daily maximum temperature exceeds the critical threshold of 37.5 °C, compared with 14% today, exposing an additional 177–246 million older adults to dangerous acute heat. Effects are most severe in Asia and Africa, which also have the lowest adaptive capacity. Our results facilitate regional heat risk assessments and inform public health decision-making.

Climate change has potentially dire consequences for the health and well-being of older adults[1,2]. Increases in the intensity, duration, and frequency of heat spells pose direct threats to physical health and mortality risk, with especially severe consequences for older adults, given their heightened susceptibility to hyperthermia and common health conditions worsened by heat exposure such as cardiovascular disease[3,4]. Older adults who are socially isolated, economically disadvantaged, have cognitive, physical, or sensory impairments, and live in substandard housing with inadequate cooling systems are especially ill-equipped to withstand or adapt to heat extremes[5,6]. Tragedies like the heat-related deaths of Florida nursing home residents following an extensive power outage during Hurricane Irma in 2017, the deaths of thousands of older adults in 21 European nations during the August 2022 heatwave[7], and the 3500 deaths - mostly among older adults - during the 2015 heatwave in India and Pakistan, highlight the threats posed by climate change-driven increases in ambient temperatures[8–10].

Despite extensive research confirming the individual-level effects of extreme heat on older adults' health and mortality risk[11], older adults' population-level heat exposure has received less attention[2,5]. Coincident trends of population aging and a warming climate portend the emergence of biologically and socially vulnerable "hotspots": countries and regions that experience both increasing concentrations of older adults and intensifying high temperature extremes[12,13]. The global population is aging at an unprecedented pace. The age 60+ population is projected to more than double by mid-21st-century - from 1.1 billion in 2021 to nearly 2.1 billion by 2050. By 2050, a projected 21% of the global population will be age 60+, with more than two-thirds of older adults residing in low- and middle-income countries where climate change-driven extreme events are especially likely[12]. Regions projected to experience the fastest growth in the relative size of their older populations and largest increases in average and maximum temperatures will experience the most rapid expansion in population-level heat exposure, and associated demands on local governments to develop appropriate infrastructures and response systems[14].

Population aging and warming both vary widely across the globe. Historically high fertility rates in developing countries primarily in the Global South have contributed to large, rapidly growing, and relatively younger populations, whereas below replacement-level fertility rates and advances in nutrition, sanitation, and biomedical innovations have contributed to rapidly aging populations in the Global North - albeit with projected long-term population declines[15]. Symmetrically, ambient temperature levels and their acceleration differ by country[16]. Local

[1]CMCC Foundation - Euro-Mediterranean Center on Climate Change, Venice, Italy. [2]RFF-CMCC European Institute on Economics and the Environment, Venice, Italy. [3]International Institute for Applied Systems Analysis, Laxenburg, Austria. [4]Department of Economics, Ca' Foscari University, Venice, Italy. [5]Department of Earth & Environment, Boston University, Boston, MA 02215, USA. [6]Department of Sociology, Boston University, Boston, MA 02215, USA. ✉e-mail: giacomo.falchetta@cmcc.it

ambient temperature distributions, and the frequency, intensity and duration of extremes[17] are heterogeneous and their future changes are uncertain due to dynamics of the warming climate[18]. Understanding the geographic overlap of these trends is thus critically important. Doing so facilitates identification of areas at extreme risk of elderly population heat exposures. Moreover, given that adaptive capacity is correlated with income, it is key to assess where exposures coincide with enhanced vulnerability due to adaptation challenges[19], e.g., purchasing and operating air conditioning[20].

Climate change impacts on heat exposure, health, and well-being as a function of contextual and individual-level characteristics have been extensively documented. Previous research has focused on climate change[21], population change[22,23], and age structure transformations[24]. Heat exposure studies have varied in geographic scope, ranging from country[25], to regional[26,27] and global assessments[21,28,29], with some focusing exclusively on urban areas[30–33]. Different metrics have been adopted to measure exposure, and potential mortality[7,34,35], including novel indicators such as "unprecedented hot summers"[26]. As well, adjacent literature has empirically assessed heat exposures' health consequences[36–38]. These studies document systematic variation in risk with socioeconomic and demographic (e.g. age, gender, race) characteristics. In addition to physical, physiological, and psychological conditions, population aging, preexisting structural inequalities in income and in health, and limited availability of and access to basic services and information are key moderators of temperature impacts on human health[39–41]. Further studies have used empirically estimated exposure-response functions to project morbidity and mortality at different geographic scales under potential demographic and climate futures[42–45]. However, relative risk estimates specifically for older adults are comparatively rare[46–48]. Within this diverse array of contributions, we are unaware of

comprehensive global-scale assessments of the future evolution of older adults' heat exposures consistent with the shared socio-economic pathway (SSP) scenarios[15].

Building on our approach developed for the United States[49], we construct global gridded age-stratified demographic projections for different population scenarios (see Methods) and combine them with temperature projections from the Coupled Model Intercomparison Project Phase 6 (CMIP6) downscaled, bias-corrected model output[50] to quantify chronic exposure to high average temperatures and the frequency and intensity of acute exposure to extreme high temperatures for different age groups across the world. Our gridded population projections are benchmarked against previous region-specific modelling work to assess their consistency (see Methods). We develop temperature exposure metrics at the same spatial resolution. For cumulative exposures we use annual cooling degree days (CDDs) above the 24° C threshold. For acute exposures metrics we use the annual count of hot days (#HDs: days with maximum temperature exceeding 37.5° C) and the 95th percentile of the 20 y daily maximum temperature distribution (TMAX95). The latter yield important insights given recent evidence that maximum temperatures are already approaching the critical thresholds of 35° C with high humidity and of 40° C with low humidity[3,51,52] in different regions of the world under current warming levels[1]. From the resulting dataset we compute the geographic intersection of current (2020) and projected mid-century (2050) changes in global and regional populations of adults age 69+, and chronic and acute heat exposures.

## Results

### Heat exposures of older individuals: global trends and distribution

Figure 1 visualises bivariate global maps of the fraction of population aged 69+, the average yearly Cooling Degree Day exposure (CDDs), the

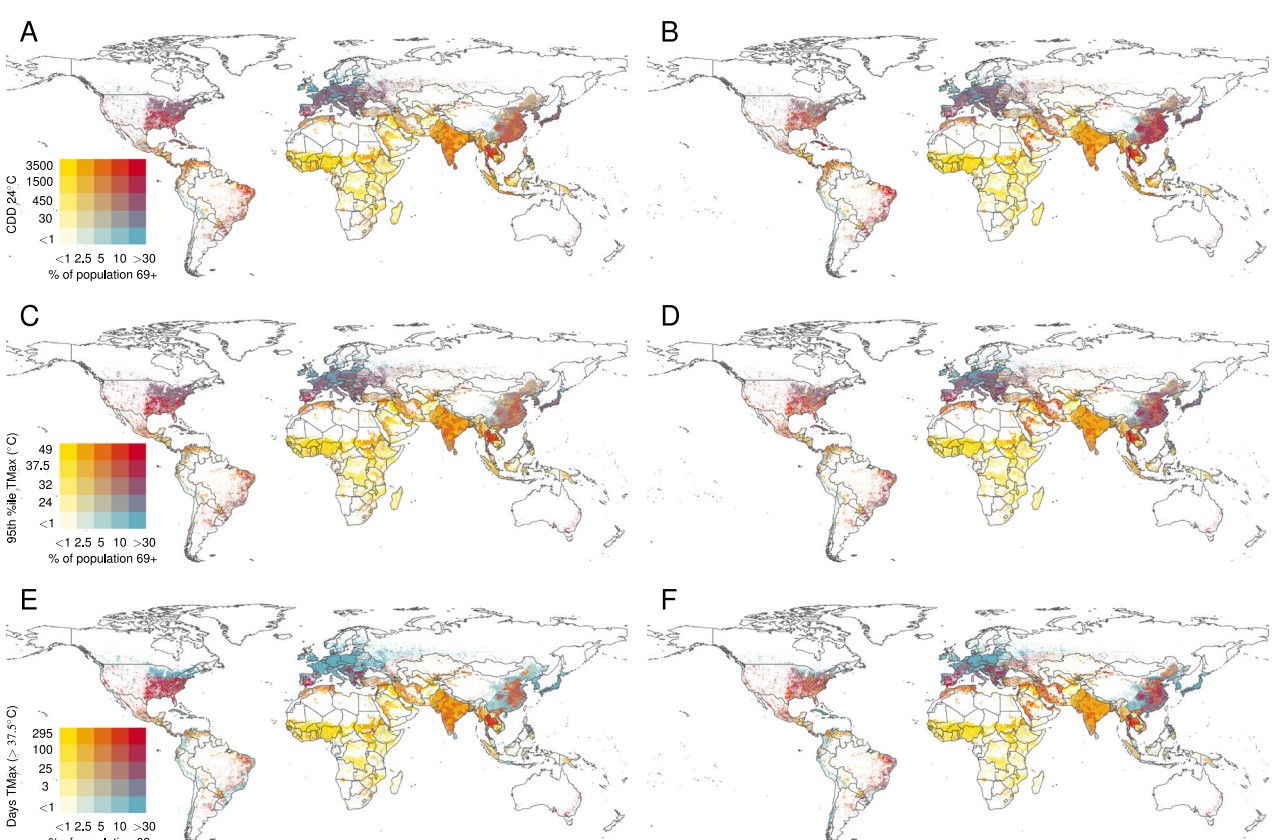

**Fig. 1 | Global intersection of aging and heat exposure in the current climate (left column) and circa 2050, SSP2(45) (right column). A**, **B** Proportion of population aged 69+ exposed to annual Cooling Degree Days (CDDs). **C**, **D** Annual temperatures corresponding to the 95th percentile of local extreme heat exposure (TMAX95). **E**, **F** Annual days with *TMAX* > 37.5° C (#HD). Figures SI-13–SI-15 in the SI Appendix present similar figures for SSP1(26), 3(70), and 5(85).

frequency of acute heat exposure defined by the number of hot days (#HDs) with daily maximum temperature above 37.5° C, and the intensity of acute heat exposure metric defined as the 95th percentile (TMAX95) of the distribution of daily maximum temperature for the combination of historical population and climatology (1995–2014; first column), and under the future evolution of the population size and structure and climate of SSP 2(45) scenario (around 2050; second column). Exposure is also projected up to around mid-21st-century along three additional climatic-demographic futures based on SSP-RCP interaction scenarios 1(26), 3(70), and 5(85) (presented in Fig. SI-13), leading to increasingly different levels of warming and demographic growth and transformations). In this scenario framework, futures are defined by the interplay between socio-demographic and greenhouse gasses radiative forcing trends (see "Methods" for a more detailed definition of the scenarios assessed in our analysis).

A north-south divide in terms of demographic composition appears in most continents, with the exception of the Americas and Oceania, where the distribution is less polarized. The maps reveal - in red - areas of growing overlapping stress in both heat exposure and an aging population (parts of the Americas, southern Europe, coastal China and several Southeast Asian countries, and Australia), as well as - in yellow and shades of orange - areas with strongly growing hot climate conditions but a relatively smaller demographic pressure because of a younger population, and, finally, areas where heat exposure is and will remain more limited but the population is rapidly aging (e.g. the northern parts of Asia and northern Europe).

To examine the results observed in the maps of Fig. 1 more closely, Fig. 2A–C displays the cumulative count of people as a function of their exposure to a given amount of CDDs, #HDs, or TMAX95 at a global scale for individuals aged >69 in the four scenarios considered. Note that solid lines represent CMIP6 General Circulation Models (GCMs) median, and light lines describe each GCM individually. These plots allow assessing the degree of spatial overlap in terms of the distribution of exposures comparing current demographics and historical climate with two SSP-RCP evolutions by the year 2050. Then, Fig. 2B–D presents population-weighted boxplots of CDDs, #HDs, and TMAX95 exposure for each global macro-region, respectively. Here the boxplot range also includes the CMIP6 GCMs range, thus incorporating the climate model uncertainty. As a benchmark, SI-16–SI-18 present similar plots for the total global population, as well as for populations aged <69, for both CDDs, #HDs, and TMAX95 exposure.

The figures show that, by the year 2050, the global heat-exposed population aged 69+ is projected to grow considerably. If thresholds of 30 hot days per year, 37.5° C, and 1200 CDDs/yr (given the lack of critical CDD-related thresholds in the literature, the value of 1200 CDDs/yr is defined as the mean number of CDDs in regions where - under a historical climate - a TMAX95 > 37.5° C is recorded) are considered as benchmark values for dangerous exposure[3] (each represented by the purple dashed vertical lines in panels A–B and E–F), then we estimate exposed population aged 69+ to increase by 0.16–0.23 billion for #HDs, 0.18–0.25 billion for TMAX95 and 0.23–0.32 billion for CDDs, respectively, depending on the scenario considered. Based on the numbers summarised in Fig. 2, we conclude that the share of the population aged 69+ will increase in all continents, reaching the largest share in Europe (representing one-fifth to almost one-quarter of the total population, depending on the SSP scenario). Similarly, also in North America it may surge up to around one-fifth of the total population. The largest absolute numbers are projected for Asia, where individuals aged 69+ will reach between 588–748 million (up to more than a threefold increase from the current 239 million).

As a result of a warmer climate and an older global population, Population Degree Days (PDDs, namely the CDD exposure of all individuals aged 69+) is projected to more than quadruple from the current 203 billion to about 778–1008 billion in 2050, with the bulk of this growth concentrated in Asia (where PDDs grow from about 150 to

585–768 billion). With regards to the frequency of hot days, globally we find a surge from the current average of 10 to 19–21 days/year, reflected by the growth in TMAX95 from 32 to about 35° degrees. This translates into population-based metrics of population at the 95th percentile (PD95, TMAX95 exposure of all individuals aged 69+) growing from 15 to 35–45 billion, determining an up to three-fold increase in the acute heat intensity exposure. Absolute numbers will be dominated by the exposed aging population in Asia, but, in relatively terms, the daily maximum temperature of the 95th percentile, TMAX95, will increase the most in Europe, going from 28 to 31° C, a 11% increment that exceeds the global average of 9%, under high warming.

In addition, Figs. SI-16–SI-18 are useful to compare the across-region and across-age groups changes in the distribution of (age 59+ population-weighted) CDDs, hot days, and °C compared to the historical climate. The first key result is that the across-region differences are visibly larger than the across-age group exposure change. The largest cumulative exposure changes are projected to occur in Africa (average elderly individual exposure growth of about 195–323 CDDs/yr), Asia (average growth of 312–397 CDDs/yr), South America (152–270 CDDs/yr) and North America (120–151 CDDs/yr).

## Age group and regional heterogeneities in heat exposures

To further investigate the underlying heterogeneities in the estimated global trends, we examine both across-region and within-region differences in heat exposure for different age groups. To quantify the absolute exposure change for each age stratum in each macro-region, Fig. 3 provides the range (with each line range representing the CMIP6 GCMs range) of the change in the number of individuals between SSP2(45) and current population based on intervals of exposure to a given level of CDDs (panel A), number of hot days #HDs (panel B), and TMAX95th (panel C). Similar Figures for the other three scenarios assessed can be found in the SI (Figs. SI-19–SI-21). In addition, Table SI-3 summarises the age-group and region-stratified count of people by CDDs exposure level in each scenario.

These results reveal that future projected changes in cumulative heat exposure are highly heterogeneous across regions, while also showing important differences between age groups - with such differences also being highly region-specific. Increases in age 69+ populations are observed in all regions, consistent with past studies of global population aging, but they are concentrated in high CDDs levels in primarily low-income nations in the Global South, concentrated in Africa, Asia, and South America. Conversely, older individuals increase in higher-income nations of the Global North including North America and Europe is more centered in low to intermediate CDD exposure levels.

When comparing exposure change across age groups within the same region and scenario, a population-weighed Student's $t$-tests of the difference in population-weighted CDD exposure reveals that in all regions (refer to Table SI-2), for both climate change scenarios, the difference in mean exposure of the <69 and 69+ age groups is statistically significant at the $p < 0.01$ level. This result suggests non-random spatial allocation of the population age groups relative to the current climate and the projected increase in cumulative heat exposure.

To complement the analysis based on cumulative exposure (measured through CDDs), Fig. 3 also illustrates the change in the absolute number of people exposed to a given acute heat level in each global region for the two age groups between years 2020 and 2050 in terms of frequency (#HDs, panel B) and intensity (TMAX95th, panel C). Increases in acute exposure are generally consistent with those expected for cumulative exposure, but they reveal additional important trends. For instance, more than 23% of the global population aged 69+ is projected to be living in climates with TMAX95 > 37.5 C° in 2050 compared to 14% in 2020, an absolute increase of about 177–246 million age 69+ individuals. Most of these populations will be concentrated in Asia and Africa. Conversely, people exposed to a TMAX95 < 37.5 C° will remain mostly concentrated in Europe and North America.

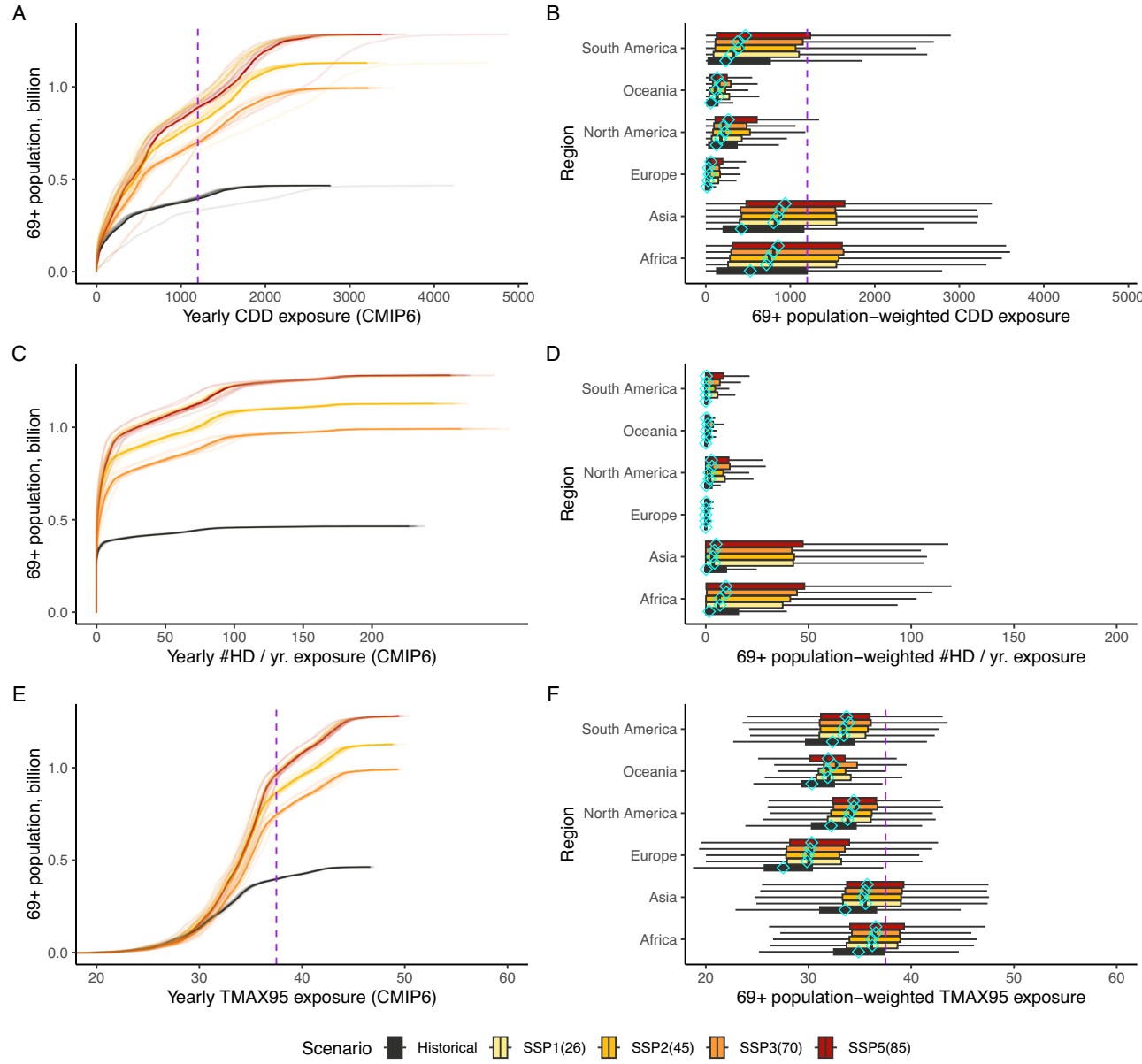

**Fig. 2 | Age 69+ heat exposures: global cumulative distributions (left column) and regional frequency distributions (right column).** Panels (**A**, **C**, **E**): cumulative counts of aged 69+ individuals worldwide exposed to a given amount of median Cooling Degree Days in a year (CDDs), the number of Hot Days, #HD, and the corresponding to 95th percentile of acute extreme heat exposure (TMAX95). Historical vs. 2050 for SSPs 1(26), 2(45), 3(70), and 5(85), CMIP6 global climate models (GCMs) (hot models[70] excluded) range; multi-model median in bold lines.

Panels (**B**, **D**, **F**): boxplots of region-specific older individuals-weighted exposure to a given amount of median CDDs, #HD, and TMAX95. Historical vs. 2050 for SSPs 1(26), 2(45), 3(70), and 5(85), CMIP6 GCMs (hot models[70] excluded). Range and multi-model median (diamond). Purple lines in panels (**A**) and (**C**) and (**E**) and (**F**) identify critical thresholds of 1200 CDDs/yr and 37.5° C (dangerous temperature even under short exposure[3]).

Such growing acute exposure is a cause for great concern - in particular for older individuals - and it will likely drive up the demand for and use of indoor thermal regulation appliances, with significant private and social costs for energy use and its externalities[53] and health repercussions for those who cannot afford them[20,54].

**Decomposing the drivers of old-age individuals' heat exposure**
The three key drivers influencing the future climate-related risks posed to human health, namely total population growth, demographic changes in the age structure, and heat exposure will evolve following context-specific, and potentially heterogeneous, trends[55]. Table 1 summarizes the future changes in those three co-occurring trends by region. Rapid population growth will occur in Africa and Asia, whereas

only a few millions will be added to the current population in Europe and South America. An already large share of age 69+ people in Europe will almost double in 2050, and this region will host about one-fifth of the global population aged 69+. A more that doubling is observed in most regions, but starting from low values, and reaching shares around 12–15%, which is close to the global average, while Africa remains well below this number. The average 95th percentile of the maximum temperature distribution increases by a at least 1° C in all regions and scenarios, but Europe shows some of the largest changes in exposure, with the average number of hot days (#HDs) increasing from 0 to 3, and number of CDDs more than tripling.

Figure 4 decomposes the overall future change in exposure ($E_r$, "Methods") in terms of Person Degree Days (panels A, B), Person Hot

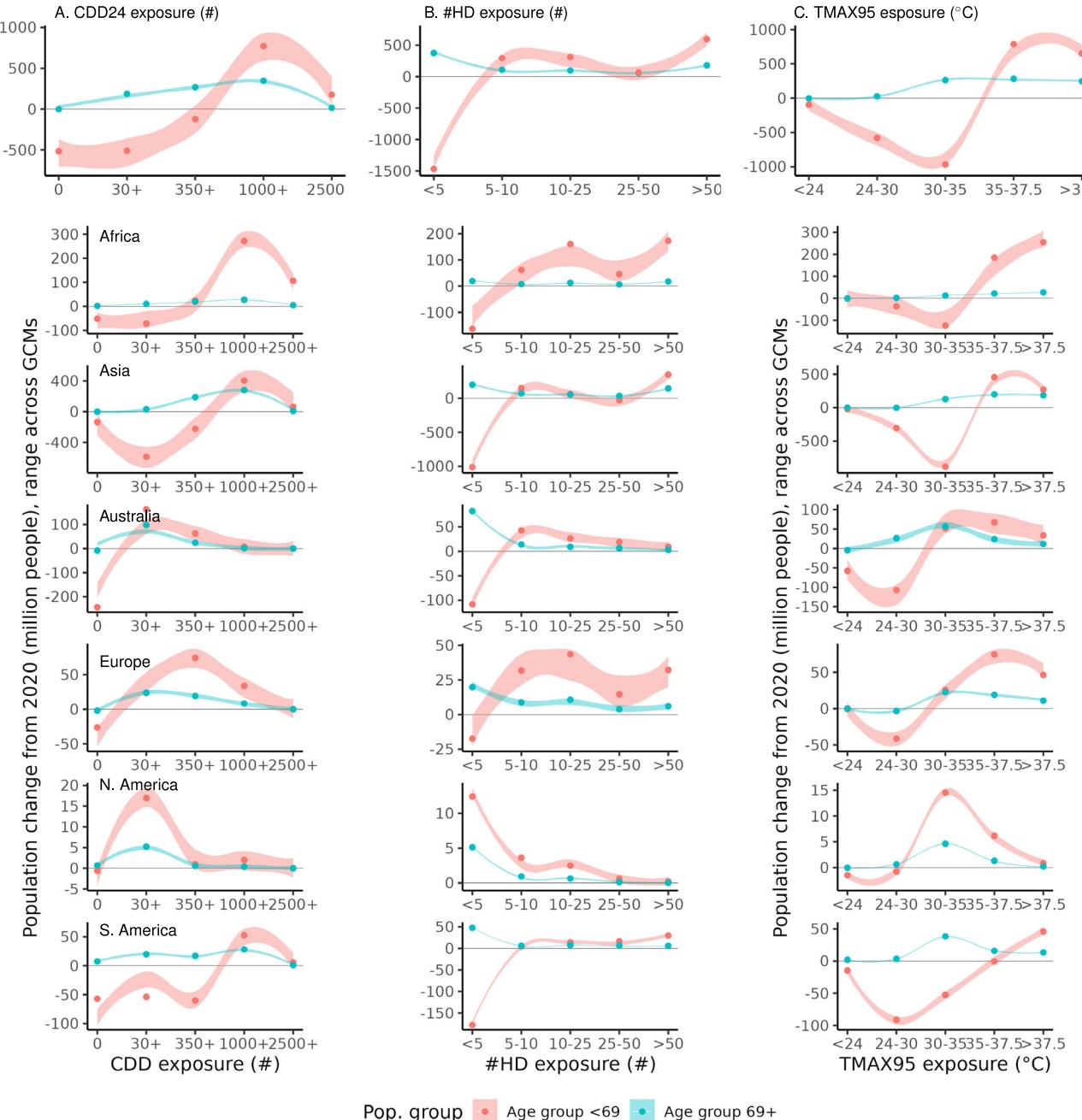

**Fig. 3 | Regional and age-group-specific trends in the cumulative and intensity of acute exposure of population groups: 2020–2050.** Top row: global trends; lower rows: regional trends. Panel (**A**): GCM uncertainty range for the count of individuals exposed to a given CDD exposure level, age stratification, faceted by region, difference between SSP2(45) and current population. Panel (**B**): GCM uncertainty range for the count of individuals exposed to a given number of annual days with TMAX95 > 37.5 °C, age stratification, faceted by region, difference between SSP2(45) and current population. Panel (**C**): GCM uncertainty range for the count of individuals exposed to a given 95th percentile maximum temperature exposure level, age stratification, faceted by region, difference between SSP2(45) and current population. Figures SI-19–SI-21 in the SI Appendix present similar figures for SSP1(26), 3(70). and 5(85).

Days (panels C, D), and Person Degrees (panels E, F) in the SSPs 1(26), 2(45), 3(70), and 5(85) scenarios across six continents. Since space aggregation in geographically large regions might hide subregional climate and socio-demographic heterogeneities, decomposition results at the country level are presented in the SI Appendix in Figs. SI-23–SI-37.

Cumulative heat exposure of older individuals (PDDs) will at least triplicate in all continents by 2050, irrespective of the scenario considered. Climate change will be the prevailing mechanism of change in more temperate nations of Europe and North America. Population aging will be the defining driver of future exposure in the warmer countries of Africa, Asia, and South America. Africa is also the region where total population growth will have the largest impact.

If absolute exposure is measured in terms of people, Asia will experience levels of older adult heat exposure nearly four times higher than the other regions put together due both to its large, aging population and hot climate. Some countries within the same macro-region (e.g. Japan and India in Asia) can show common large increases in total heat exposure, despite great differences in their population structure in the 1994–2014 period (see Figs. SI-23–SI-37). Older adults cumulative heat exposure in other regions will also grow robustly, despite being significantly smaller in absolute terms.

**Table 1 | Evolution of older heat exposure determinants under historical climate and current demographics, and under scenarios SSP2(45) and 5(85). CDDs, TMAX95, #HDs, PDDs, PHDs, PD95th report the median and the interquartile range in brackets**

| | Population (10⁶) | 69+ (%) | CDD24 # | PDDs (10⁹) | TMAX95 (°C) | PD95s (10⁶) | #HDs # | PHDs (10⁶) |
|---|---|---|---|---|---|---|---|---|
| **Historical climate** | | | | | | | | |
| Africa | 1347 | 2.1 | 684 | 19 | 35.2 | 996 | 25 | 697 |
| | | | [671, 693] | [19, 20] | [35.1, 35.3] | [994, 999] | [24, 25] | [695, 708] |
| Asia | 4517 | 5.3 | 630 | 150 | 34.1 | 8114 | 15 | 3533 |
| | | | [624, 642] | [148, 153] | [33.9, 34.2] | [8075, 8127] | [14, 15] | [3426, 3558] |
| Australia | 36 | 8.6 | 109 | 0 | 30 | 94 | 2 | 5 |
| | | | [107, 117] | [0, 0] | [29.9, 30.1] | [93, 94] | [1, 2] | [4, 5] |
| Europe | 860 | 12.3 | 36 | 4 | 27.8 | 2935 | 0 | 51 |
| | | | [34, 40] | [4, 4] | [27.8, 27.9] | [2934, 2939] | [0, 0] | [48, 53] |
| N. America | 372 | 11.2 | 213 | 9 | 32.5 | 1352 | 5 | 202 |
| | | | [198, 222] | [8, 9] | [32.4, 32.6] | [1349, 1357] | [5, 5] | [191, 212] |
| S. America | 680 | 5.6 | 414 | 16 | 31.8 | 1203 | 3 | 123 |
| | | | [410, 425] | [16, 16] | [31.6, 31.9] | [1198, 1206] | [3, 4] | [113, 131] |
| World | 7967 | 5.8 | 437 | 203 | 32.3 | 15025 | 10 | 4632 |
| | | | [431, 444] | [200, 206] | [32.2, 32.4] | [14,979, 15,050] | [10, 10] | [4476, 4701] |
| **2050 SSP245** | | | | | | | | |
| Africa | 1943 | 3.9 | 907 | 69 | 36.5 | 2768 | 37 | 2810 |
| | | | [864, 942] | [66, 71] | [36.3, 36.7] | [2754, 2779] | [34, 37] | [2546, 2829] |
| Asia | 4951 | 13.4 | 942 | 627 | 36.1 | 23,984 | 25 | 16,824 |
| | | | [918, 975] | [611, 648] | [36, 36.4] | [23,930, 24,192] | [25, 26] | [16,335, 17,539] |
| Australia | 55 | 15.5 | 182 | 2 | 31.4 | 268 | 3 | 24 |
| | | | [159, 189] | [1, 2] | [31.2, 31.7] | [267, 271] | [2, 3] | [21, 27] |
| Europe | 886 | 21.5 | 104 | 20 | 30.4 | 5800 | 3 | 506 |
| | | | [94, 114] | [18, 22] | [30.1, 30.5] | [5741, 5817] | [2, 3] | [447, 585] |
| N. America | 443 | 17.4 | 326 | 25 | 34.3 | 2642 | 11 | 828 |
| | | | [309, 351] | [24, 27] | [34, 34.5] | [2626, 2659] | [9, 11] | [680, 868] |
| S. America | 730 | 14.1 | 566 | 58 | 32.9 | 3392 | 7 | 752 |
| | | | [554, 610] | [57, 63] | [32.8, 33.1] | [3379, 3411] | [7, 8] | [677, 825] |
| World | 9064 | 12.4 | 716 | 808 | 34.7 | 39,106 | 19 | 21,756 |
| | | | [692, 744] | [781, 840] | [34.5, 34.9] | [38,968, 39,393] | [19, 20] | [21,063, 22,492] |
| **2050 SSP585** | | | | | | | | |
| Africa | 1689 | 5.5 | 992 | 91 | 36.9 | 3399 | 40 | 3690 |
| | | | [967, 1053] | [89, 97] | [36.7, 37] | [3382, 3414] | [38, 41] | [3504, 3785] |
| Asia | 4547 | 16.4 | 1027 | 768 | 36.5 | 27,273 | 28 | 20,661 |
| | | | [1004, 1088] | [750, 813] | [36.3, 36.6] | [27,128, 27,405] | [26, 28] | [19,513, 21,199] |
| Australia | 62 | 16 | 192 | 2 | 31.7 | 315 | 3 | 30 |
| | | | [185, 207] | [2, 2] | [31.6, 31.8] | [315, 316] | [3, 3] | [28, 32] |
| Europe | 963 | 22.9 | 130 | 29 | 31 | 6830 | 4 | 816 |
| | | | [111, 141] | [24, 31] | [30.3, 31.2] | [6674, 6874] | [3, 4] | [677, 974] |
| N. America | 526 | 17.3 | 368 | 34 | 34.6 | 3139 | 12 | 1095 |
| | | | [342, 405] | [31, 37] | [34.4, 35] | [3125, 3174] | [11, 13] | [1013, 1199] |
| S. America | 641 | 17.4 | 652 | 73 | 33.4 | 3716 | 9 | 1010 |
| | | | [624, 691] | [70, 77] | [33.2, 33.6] | [3696, 3742] | [8, 10] | [921, 1137] |
| World | 8485 | 15.1 | 786 | 1008 | 35.1 | 44,999 | 21 | 27,075 |
| | | | [752, 826] | [964, 1058] | [34.8, 35.3] | [44,651, 45,183] | [21, 22] | [26,401, 28,206] |

Table SI-1 reports figures for scenarios SSP1(26) and 3(70).

Moreover, we observe that climate change has a strong impact on the frequency of acute exposure (Panels C, D), but a relatively negligible impact on the burden of the intensity of acute exposures (Panels E, F), which is mostly affected by the shift in the age structure of population in areas where extreme temperatures are already high.

## Discussion

Our projected acceleration of both chronic and acute heat exposures of older adults raises a serious concern for global public health[56–58], given older adults' reduced capacity to thermoregulate, their greater number of co-morbidities, and their reliance on medications that cause dehydration[59,60]. Additionally, older adults with cognitive or

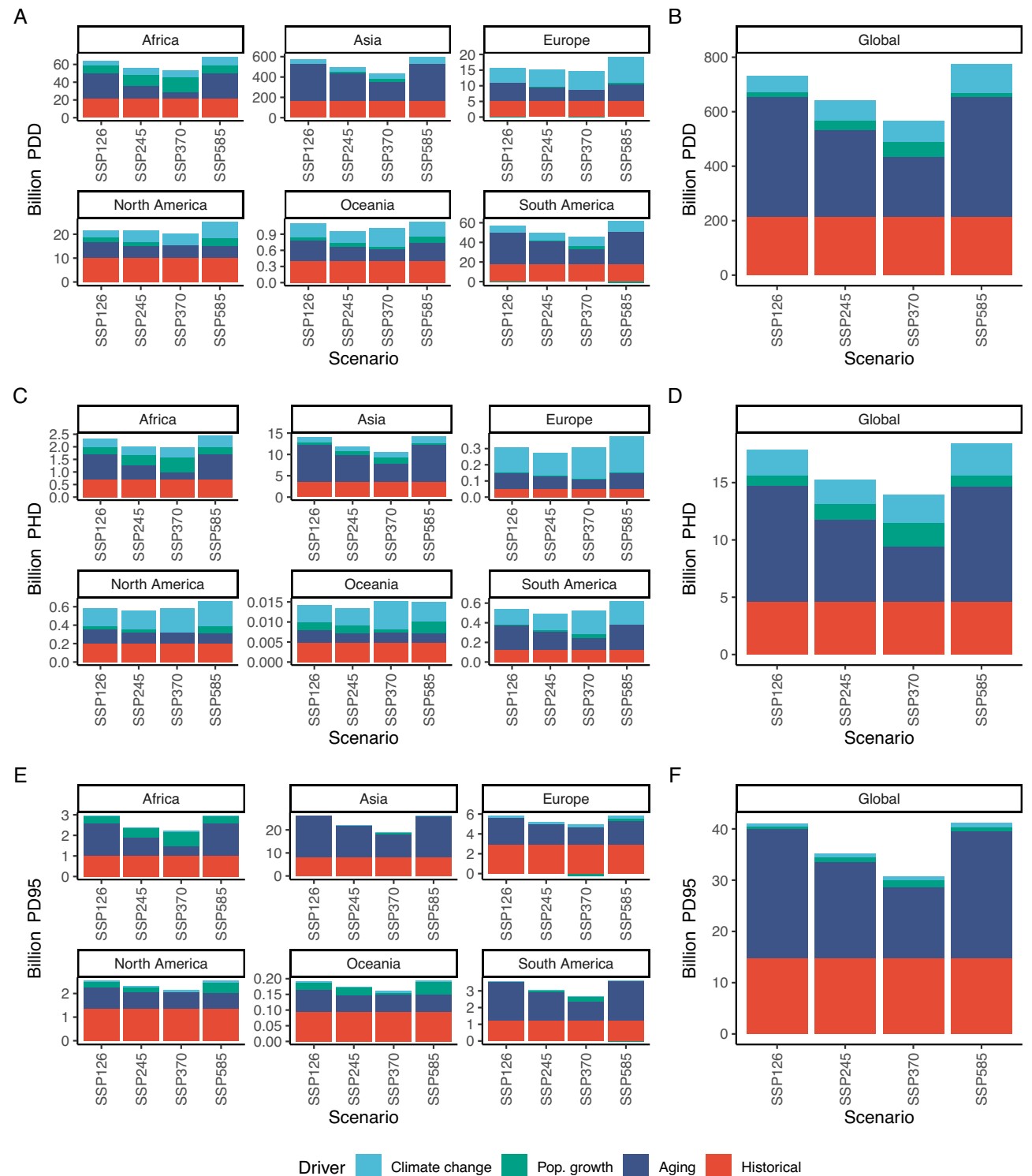

**Fig. 4 | Decomposition of projected determinants of exposure by region and SSP scenario.** Panel (**A**, **B**): Billion Person Degree Days (PDDs) decomposition, by region and global. Panel (**C**, **D**): Billion Person Hot Days (PHDs) decomposition, by region and global. Panel (**E**, **F**): Billion Person Degrees (PD95th) decomposition, by region and global.

physical limitations, insufficient housing or economic resources, and no geographically distant family are poorly equipped to adapt to heat extremes. Areas with aging populations and rising heat exposures are thus likely to face considerable demands for social and health services, requiring novel policy interventions[49].

The coming decade will be critical for the agendas of both climate change and healthy ageing[12]. There is a pressing need for stakeholders

in both agendas to both understand and work to address the interconnections between them. In particular, the two dimensions should be integrated into adaptation planning and healthcare directives in order to minimize intensifying heat's direct morbidity and mortality impacts, and their indirect costs on society more broadly. Potential interventions abound, including increasing the penetration of active (e.g., air conditioning) and passive cooling technologies in buildings[61],

**Table 2 | Demographic input data sources and characteristics**

| Data source | Geography | Period | Description |
|---|---|---|---|
| Gridded, age-stratified, global population | 100 m | 2020 | Top-down constrained age/sex structure estimate[73]. |
| Downscaled global population projections | 0.00833° | 2020–2100, decadal | Gridded downscaled population projections, five SSP scenarios[76]. |
| Country-level, age-stratified, global population projections | Countries | 2020–2100, decadal | Population by age group, gender and level of education, five SSP scenarios[15]. |
| Regional, age-stratified, US population projections | County | 2020–2100, decadal | Modeled demographic and migration dynamics to project future age-stratified population scenarios, five SSP scenarios[77]. |
| Regional, age-stratified, European population projections | EU NUTS-3 | 2020–2100, decadal | Modeled demographic and migration dynamics to project future age-stratified population scenarios, five SSP scenarios.[79] |

increasing building albedo and/or enlarging green spaces and tree canopy cover to counteract climate change-driven enhancement of the urban heat island effect[62], expanding heat early warning systems[63] and providing accessible public cooling[64]. Quantifying both the costs of these alternatives, and their effectiveness in terms of moderating morbidity and mortality risk, is an area of research that is ripe for investigation[53,65]. However, the aforementioned measures do not specifically address the distinctive needs of large and rapidly growing populations of older adults, which our results emphasize should be explicitly considered. Many 69+ individuals are disproportionately vulnerable to the effects of heat extremes due to chronic health conditions and illnesses, physical, sensory, or cognitive disabilities, and social isolation, marginalization, and lacking access to healthcare or resources for private adaptation. Incorporating these additional concerns into the design and deployment of adaptation solutions will likely pose significant challenges.

Notwithstanding this paper's data and methodological innovations, the future size and spatial distribution of elderly populations remain uncertain. Projection techniques and spatial downscaling approaches invariably necessitate assumptions, which have their own limitations. Perhaps the most significant structural impediment is the lack of harmonized global-scale high-resolution age-stratified population counts that can serve as a reliable initial condition on which to base future sub-national projections. Holding this problem aside, a further limitation is that our results may overestimate actual exposures given that some amount of autonomous adaptation is likely. Well-resourced older adults are likely to undertake costly private actions to adapt (e.g., shielding themselves through purchasing and operating air conditioners). Moreover, beyond individual decision-making, well-resourced communities may find it optimal to invest in one or more of the interventions described above. Given these possibilities, the deeper question of what should be considered a baseline scenario in projections is fraught with uncertainty. Combining our estimates with other datasets, e.g. urban green space cooling potential[66] or air conditioning availability[61], can facilitate the elaboration of future potential net exposures. But exploration of these feedbacks will need to take into account the structural evolution of patterns of mobility, outdoor time allocation, appliance adoption and use, household energy demand, and electric power load duration curves in response to societies' differential patterns of aging[67]. Finally, additional phenomena, such as climate-induced migration, may itself have direct impacts on demographic change within and across regions[68].

The output data of our analysis are made publicly available in a repository and can be beneficial for health-related assessments and adaptation planning. Future research can leverage these data to inform decision-making and to be used in risk and mortality studies, as well as to support future sub-national, age-stratified population projection studies.

## Methods
### Data
**Current and future daily temperatures.** We use daily temperatures on a global 0.25° grid from the NASA Earth Exchange Global Daily Downscaled

Projections (NEX-GDDP-CMIP6) dataset[69]. These data are downscaled outputs from the CMIP6 ScenarioMIP exercise's bias-corrected runs of 14 GCMs (the CMIP6 GCMs excluded of the 'hot models'[70]) for the 1995–2014 historical period, as well as future projections for the 2041–2060 period under four scenarios that combine the Shared Socioeconomic Pathway (SSP) and the Representative Concentration Pathway (RCP) assumptions[50]: SSP126 (a combination of the SSP1 and the ambitious emissions mitigation RCP 1.6 scenario); SSP245 (a combination of the SSP2 "middle-of-the-road" continuation of historical trends[71] and the moderate-warming RCP 4.5 scenario); SSP370 (a combination of the SSP3 (high challenges) and the high emissions RCP 7.0 scenario); and SSP585 (a combination of the SSP5 "fossil-fueled development"[72] and RCP 8.5 very high-warming scenarios).

**Population datasets.** Age-stratified gridded population counts are derived from WorldPop[73] and Pezzuolo et al.[74] on a 1-km grid for the year 2020. We aggregate age strata in these data into two broad categories, <69 and 69+. The choice of the 69-year-old threshold is justified on grounds that - as reported in recent governmental reports[75] - given changes in remaining life expectancy over time, age 70 can be thought of as the "new age 65" in terms of health.

In addition, we consider downscaled SSP-consistent total population projections[76] and country-level SSP-consistent age-stratified population projections[15]. The population data considered in the analysis are summarised in Table 2.

Age-stratified population projections are already available only for a limited number of regions (and for each regions projections are calculated with different methodologies and underlying assumptions), and therefore a consistent imputation method is used for our global projections. Next section describes the methodology and its validity and how the resulting projections compare with previous existing age-stratified, sub-national demographic projections to 2050.

**Software implementation.** The analysis is carried out in the R scientific computing environment, mainly relying on *terra*, *raster*, *sf*, and *tidyverse* packages. The open-source code for replication is found on Github under https://doi.org/10.5281/zenodo.10417641, listing additional software and data requirements for replication. Moreover, the software implementation of the mathematical age-stratified population downscaling and projection procedure is exemplified by an R script, reported in the SI of the paper.

### Gridded population growth and age 69+ population fraction circa mid-21st-century
**Construction.** Let $a$, $c$ and $s$ index age strata, countries and SSP scenarios, respectively. Our starting point is WorldPop unconstrained gridded population estimates for the 2020 base year[73], aggregated to 1 km resolution for each grid cell, $g$: $n_{a,g,0}$. The objective is to estimate the year-2050 gridded population by age for each SSP scenario:

$$N^*_{a,g,s} = \gamma^*_{a,g,s} \times n_{a,g,0} \tag{1}$$

where $\gamma^*$ denotes unknown growth factors by age-stratum ($a$), by grid cell ($g$) and scenario ($s$) that must be imputed. We proceed by assuming that WorldPop is an accurate representation of the current population. The population growth factors consistent with gridded projections of the total mid-21st-century population prepared by Gao[76] ($N^{Gao}$) are:

$$\gamma^T_{g,s} = \frac{N^{Gao}_{g,s}}{\sum_a n_{a,g,0}} \qquad (2)$$

We can then compute age-, scenario-specific average growth factors at the country level that modify the total population grid-cell level growth factors for each stratum, $\lambda$. Specifically, we want the grid cell-level population in each stratum to add up to country-level population projections by age ($N^{KC-Lutz}$) by KC and Lutz[15], where $g(c)$ is a mapping that identifies the subset of grid cells that belong to country $c$:

$$N^{KC-Lutz}_{a,c,s} = \lambda_{a,c,s} \times \sum_{g(c)}(\gamma^T_{g,s} \times n_{a,g,0}) \Rightarrow \lambda_{a,c,s} = \frac{N^{KC-Lutz}_{a,c,s}}{\sum_{g(c)}(\gamma^T_{g,s} \times n_{a,g,0})} \qquad (3)$$

The desired growth factors by grid cell and age stratum for each country are thus given by:

$$\gamma^*_{a,g(c),s} = \lambda_{a,c,s}\gamma^T_{g,s} \qquad (4)$$

from which we impute the gridded future age population fractions above and below the 69 y.o. cutoff as:

$$A^+_{g,s} = \sum_{a \geq 69} N^*_{a,g,s} / \sum_a N^*_{a,g,s} \qquad (5a)$$

$$A^-_{g,s} = \sum_{a < 69} N^*_{a,g,s} / \sum_a N^*_{a,g,s} \qquad (5b)$$

The limitation of this approach is its assumption of geographically homogeneous, within-country trends by age stratum.

**Comparison of downscaled elderly population data with previous projection studies.** To compare the results of our grid-cell level age-stratified demographic projections downscaling to 2050, we carry out a comparative analysis for a set of regions where demographic projection studies have been produced and made available at a sub-national spatial resolution. This exercise is based on comparing such existing projections with the 2020 individuals aged >69 counts obtained with our global method described above at each sub-national unit. Projection datasets used for this comparison include two distinct sources for the United States of America[77,78] the European Union[79], India[80], the United Kingdom[81], and China[82]. These datasets were generated through region-specific modelling, embedding country-specific assumptions or scenarios, and they are available at different sub-national unit-scale for each country and study, spanning from the US county level (2997 units) for the USA, to the state and union territory level for India (30 units), the Local Authority Districts level for the UK (301 units), the province level for China (31 units), and the NUTS-3 level for the European Union (1168 units). An important remark is that while some of these studies specifically refer to the SSP framework and its scenarios and therefore are in principle directly comparable with our gridded projections, other such as the EU projections are not based on any specific SSP. Thus, in particular for the case of the EU, part of the estimated discrepancies are likely owing to different underlying scenario assumptions (e.g., for the EU see the relevant technical note https://ec.europa.eu/eurostat/cache/metadata/Annexes/proj_esms_an1.png). The exercise is based on comparing at each sub-national unit age-stratified sub-national

population projections from the datasets detailed above with the 2050 individuals aged >69 gridded projected counts obtained with the global method described above. To evaluate the absolute and relative difference between the two data sources, we calculate metrics of *APE* and *MAPE* (Absolute Percentage Error and Mean Absolute Percentage Error), besides reporting the total absolute error at different levels of spatial aggregation.

The results of the comparison exercise (see Figs. SI-1–SI-12 for complete results) are summarised in Table 3, reporting comparisons for different countries and scenarios under different benchmarking metrics. Note that the table also specifies whether the reference scenario to which our projections are compared to is based on SSP assumptions, too, or instead it relies on its own scenario assumptions. Also note that in the table we report both metrics that summarise country-level (total) difference, and sub-national unit-level mean difference. The figures show that for the United States our projected elderly population in 2050 closely aligns with estimates from ref. 77, while it has a more significant bias in relation to the estimates of ref. 78 (irrespective of both studies being at the county level). For the European Union NUTS3 units projections and the United Kingdom local authorities projections (which are both not SSP-consistent), we observe that our SSP3 projections lead to very similar results to previous estimates from ref. 79 and ref. 81, while the other SSPs diverge more significantly. For China, where projections by ref. 21 are SSP-consistent, we observe a more significant absolute difference for all scenarios, which, however, given the very large population size of the country, results in moderate total and local percentage errors. Finally, for India, for which only SSP2 is currently available from ref. 80, we find evidence of a relatively good consistency.

**Population heat exposures**
We consider three meteorological indicators that vary by SSP scenario. The first is cooling degree days (*CDD*), a measure of individuals' cumulative (chronic) heat exposure. The second is the 20-year average 95th percentile of daily maximum temperatures (*TMax95th*), a measure of the intensity of individuals' acute exposure to heat extremes. The third is the 20-year average count of days with a dry bulb maximum temperature of 37.5° C (*#HD*), which indicates the frequency of acute "extreme hot days" exposure.

The calculations in this subsection apply to current and future climate epochs/SSP scenarios. Accordingly, we suppress the SSP scenario subscript to reduce notational clutter. CDDs for each year $t$ and grid cell $g$ are computed based on a temperature threshold, $T^* = 24°$ C, as the difference between diurnal average temperature, $\overline{T}_{g,d}$, and the threshold, cumulated over the days, $d$, in each year:

$$CDD_{g,t} = \sum_{d(t)} \max[0, \overline{T}_{g,d(t)} - T^*] \qquad (6a)$$

The variable TMAX95 is calculated based on each grid cell's 20-year distribution of daily maximum temperatures, $T^{max}_{g,d(t)}$:

$$TMAX95_g = \mathscr{P}[\{T^{max}_{g,d(t)}\}, 95] \qquad (6b)$$

The variable *#HD* is calculated based on each grid cell's 20y distribution of daily maximum temperatures, considering a threshold maximum temperature *TMAX** = 37.5° C[3] to classify a day as "hot":

$$\#HD_g = \sum_t d_g(t \in 20y)[TMAX_{g,d(t)} \geq TMAX^*] \qquad (6c)$$

Given the grid-cell population, $N_g$, and the older adults fraction from (5), $A_g$, the meteorological indicators in eq. (6) yield three heat population exposure metrics: cumulative population degree days (*PDD*), acute population degrees at the 95th percentile (*PD95*), and

**Table 3 | Summary metrics of the age-stratified population projection comparison exercise**

| Area | Source | Comparison scenario | Spatial unit | Total error ($10^6$ people 69+) | Percent error | Mean local percent error |
|------|--------|---------------------|--------------|--------------------------------|---------------|--------------------------|
| United States | 77 | SSP1 | Counties | 1.7 | 2.3 | 22.5 |
| United States | 77 | SSP2 | Counties | 0.7 | 1.1 | 22.8 |
| United States | 77 | SSP3 | Counties | 1.1 | 1.6 | 23.2 |
| United States | 77 | SSP5 | Counties | 0.5 | 0.7 | 23.0 |
| United States | 78 | SSP1 | Counties | 3.5 | 4.3 | 70.2 |
| United States | 78 | SSP2 | Counties | 1.9 | 2.7 | 70.2 |
| United States | 78 | SSP3 | Counties | 1.7 | 2.9 | 59.7 |
| United States | 78 | SSP5 | Counties | 1.9 | 2.3 | 80.8 |
| European Union | 79 | SSP1 | NUTS3 | 29.4 | 28.6 | 37.6 |
| European Union | 79 | SSP2 | NUTS3 | 14.6 | 14.2 | 31.5 |
| European Union | 79 | SSP3 | NUTS3 | 2.5 | 2.5 | 28.4 |
| European Union | 79 | SSP5 | NUTS3 | 32.3 | 31.4 | 38.8 |
| United Kingdom | 81 | SSP1 | Local authorities | 3.2 | 29.3 | 57.8 |
| United Kingdom | 81 | SSP2 | Local authorities | 1.6 | 14.4 | 43.2 |
| United Kingdom | 81 | SSP3 | Local authorities | 0.3 | 2.5 | 36.6 |
| United Kingdom | 81 | SSP5 | Local authorities | 3.5 | 32.4 | 58.8 |
| China | 82 | SSP1 | Provinces | 44.8 | 13.1 | 19.5 |
| China | 82 | SSP2 | Provinces | 56.4 | 17.7 | 22.1 |
| China | 82 | SSP3 | Provinces | 64.9 | 22.2 | 25.6 |
| China | 82 | SSP5 | Provinces | 47.4 | 13.7 | 19.2 |
| India | 15 | SSP2 | Regions | 12.7 | 7.1 | 24.9 |

population hot days frequency (*PHD*):

$$PDD_g = A_g^+ \times N_g \times CDD_g \tag{7a}$$

$$PD95_g = A_g^+ \times N_g \times T95_g \tag{7b}$$

$$PHD_g = A_g^+ \times N_g \times \#HD_g \tag{7c}$$

Using grid cells as the fundamental unit of analysis, eq. (7) forms the basis for Figs. 1 and 2 panels A, C and E. Figure 2 panels B, D and F, as well as Figs. 3 and 4 on regional geographies necessitates aggregation over grid cells of exposure, population, and meteorology. Let *H* denote climate change-driven increases in heat in (6), and *E* denote the population exposures in eq. (7). Generalizing the foregoing results to the scale of regions, *r*, population exposure of older individuals in any current or future climate scenario, $E_{r,s}$, is simply the sum over grid cells of eq. (7):

$$E_{r,s} = \sum_{g \in r} \left( A_{g,s}^+ \times N_{g,s} \times H_{g,s} \right) \tag{8}$$

In Fig. 3 we compute the regional sum of gridded elderly and non-elderly population, and the population-weighted average of gridded meteorological indexes, where the weights are stratified by age according to eq. (5). On the significance of population weighting in climate exposure metrics, see[13].

**Decomposing exposure change into the contributions of its drivers**

At the grid cell level, the fractional changes in the meteorological indicators and in the associated population exposures between the current and potential future climates can be expressed as:

$$\frac{\Delta H_{g,s}}{H_{g,0}} = \begin{cases} (CDD_{g,s} - CDD_{g,0})/CDD_{g,0} & \text{Chronic/cumulative} \\ (T95_{g,s} - T95_{g,0})/T95_{g,0} & \text{Acute,intensity} \\ (\#HD_{g,s} - \#HD_{g,0})/\#HD_{g,0} & \text{Acute,frequency} \end{cases} \tag{9}$$

$$\frac{\Delta E_{g,s}}{E_{g,0}} = \begin{cases} (PDD_{g,s} - PDD_{g,0})/PDD_{g,0} & \text{Chronic/cumulative} \\ (PD95_{g,s} - PD95_{g,0})/PD95_{g,0} & \text{Acute,intensity} \\ (PHD_{g,s} - PPHD_{g,0})/PPHD_{g,0} & \text{Acute,frequency} \end{cases} \tag{10}$$

where, as before, the index 0 indicates the current period. Algebraically decomposing the fractional change in regional-scale exposure yields the sum of three terms that capture the weighted averages of grid-cell level fractional shifts in age structure, population size, and meteorology:

$$\frac{\Delta E_{r,s}}{E_{r,0}} = \varepsilon_{r,s}^A + \varepsilon_{r,s}^N + \varepsilon_{r,s}^H = \left( \sum_{g \in r} w_{g,s} \frac{\Delta A_{g,s}^+}{A_{g,0}^+} \right) + \left( \sum_{g \in r} w_{g,s} \frac{\Delta N_{g,s}}{N_{g,0}} \right) + \left( \sum_{g \in r} w_{g,s} \frac{\Delta H_{g,s}}{H_{g,0}} \right) \tag{11}$$

in which the weights are grid cells' fractional contributions to total exposure, $w_{g \in r,s} = A_{g,s} N_{g,s} H_{g,s}/E_{r,0}$.

**Reporting summary**

Further information on research design is available in the Nature Portfolio Reporting Summary linked to this article.

## Data availability

The data generated in this study are deposited in the Zenodo database under accession code 8409700 https://doi.org/10.5281/zenodo.8409700. The input age-stratified gridded population data used in this study are available from WorldPop https://hub.worldpop.org/

geodata/summary?id=24798. The input gridded population projections data used in this study are available from NASA SEDAC https://doi.org/10.7927/q7z9-9r69. The input country-level age-stratified population projection data used in this study are available from the IIASA SSP Database https://tntcat.iiasa.ac.at/SspDb. The input downscaled NEX-GDDP-CMIP6 climate change projections data used in this study are available from the NCCS THREDDS Data Catalog https://ds.nccs.nasa.gov/thredds/catalog/AMES/NEX/GDDP-CMIP6/catalog.html.

## Code availability

The replication code, inclusive of instructions for accessing the input data and producing the output data generated in this study are available at the following Github repository: https://doi.org/10.5281/zenodo.10417641.

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

## Acknowledgements
G.F. and E.D.C. were supported by the ENERGYA project, funded by the European Research Council (ERC), under the European Union's Horizon 2020 research and innovation program, through grant agreement No. 756194. I.S.W. was supported by the U.S. Department of Energy, Office of Science, Biological and Environmental Research Program, Earth and Environmental Systems Modeling, MultiSector Dynamics under Cooperative Agreements DE-SC0016162 and DE-SC0022141.

## Author contributions
G.F. and I.S.W. conceived the experiments and designed the methodology; I.S.W. processed the climate data; G.F. conducted the experiments and generated the figures. G.F., I.S.W., E.D.C. and D.C. analysed the results and wrote the manuscript.

## Competing interests
The authors declare no competing interests.
