## [Peer Review File · Nature Communications]

Aging in a warming world: global projections of heat exposure of older adultsReviewers' Comments:

Reviewer #1:

Remarks to the Author:

This paper combined the recent CMIP6 climate projections with downscaled population aging projection to estimate the future heat exposure for different age groups using three different metrics. It also provides a detailed decomposition of the drivers of old-age individuals' heat exposure, considering three co-occurring trends: future heat exposure, overall population growth, and demographic changes in the age structure. Overall, this paper addresses a much-needed topic on the amplifying role of population aging on heat exposure under climate change. I have the following comments for the authors to consider.

1. A major question is that despite the usage of three different metrics, the results look quite similar and all metrics convey quite the same message. It's unclear to the reviewer why presenting all three metrics throughout the main text? A suggestion is to keep one as the main analysis and put others in the supplementary (or just one main figure showing the similarity for results from different metrics).
2. Another question is regarding the choice of the Shared Socioeconomic Pathway (SSP) scenarios. This study uses only two Shared Socioeconomic Pathway (SSP) scenarios – SSP245 and SSP585. The SSP585 per se is not regarded as the 'business as usual' scenario any more in CMIP6 simulations, due to the existing climate policies. Given the availability of the SSP126 and SSP370 scenarios in the NEX-GDDP-CMIP6 datasets, it is unclear why the authors only chose two of them. The inclusion of more scenarios could have provided a more comprehensive understanding of the potential impacts of climate change on the aging population.
3. This may be the most critical concern. The conclusion of the amplifying role of population aging on the heat exposure under climate change at the global scale is already covered in many previous studies, such as the Lancet Countdown Report on Health and Climate Change. Thus, the novelty of this study lies in the usage of the downscaled population aging projections under SSP scenarios in estimating future heat exposure, the validity of which is a major concern. Given the availability of age-stratified population projections is limited for some regions, validation of the imputation method used in this study needs further discussion. To understand the accuracy of this imputation method in the US and Europe in Figure 5 (i.e., understand the MAPE range), it is suggested to present the US and Europe specific population aging maps to double check the spatial pattern of population aging. The Striessnig et al. paper emphasized the importance of within country spatial variations of the population aging trends under different SSPs. If the 1-km downscaled population aging projection data is acceptable, it will have huge implications for many researches in the field and thus warrants further careful examination.
4. In calculating Cooling Degree Days (CDD), why using daily max temperature as the metric and why 37.5 degree C as the threshold? Other papers have used other metrics such as daily mean temperature above 18.3 degree C as the definition of CDD (e.g., this Nature Sustainability paper: <https://www.nature.com/articles/s41893-019-0441-9>)

Reviewer #2:

Remarks to the Author:

Aging in a warming world: global projections of cumulative and acute heat exposure of older adults

This article examined future climate change risks in world scale. They considered climate, demographic and socioeconomic scenarios in projection of heat exposure. The results highlighted aging society in Africa and Asia are most vulnerable to future climate change. In my opinion, this article may include important contents that may be interesting to some readers, but this article is not novel at all and poorly explained about their results with other data. In addition, I do not think authors know previous articles which already studied important factor such as demographic changes and adaptation using valuable statistic methods. Thus, I recommend to reject this paper.

Specific comments:

1. In introduction part, authors need to read and refer more previous projection studies. Based on my knowledge, there are more projection articles related health effects under RCP, SSP scenarios in important regional scale, such as North America, Europe area, Asia (around China) and etc. They already dealt with contents addressed in this articles. Authors need to find them, introduce some of them (most related articles to this) briefly and show difference between them and your work in the introduction part.

2. It is interesting that authors use CDD and HDs. Why do you use this term?

3. In method section, is there any logical reason you choose 69+ as elderly population? Please explain.

4. In method section, authors mentioned that age-stratified population projection are available for US and Europe. Thus they use that data for rest of the world. But this is not correct. You can find future age structure data for each country in UN website. Please find more available data.

As you already know, age structure in Asia is far different from Europe and USA. So, applying same strategy in analysis is unreasonable and not logical here.

5. As you presented in Figure 4, major factor here is aging. Previous studies mentioned about the importance of aging in climate change (please find "Projection of future temperature-related mortality due to climate and demographic changes" in EI journal). I do not think authors know about this.

6. Except factors author mentioned here, other socioeconomic factors affect future health effect in climate change. You can find more data about that and can apply in every country. This model you used here is too simple now.

7. Exposure is not always related to health effects. The degree of effects will be different based on each country's developments and adaptation strategies. Like mentioned in #6 comments, there are specific data (insurance, electricity usage...) which related to exposure-health correlation. Please address that or at least mention that in limitation.

8. Did you use R software for calculation? Add the sentence what software you used for your analysis if there is any.

9. Figure presentations: Figures you showed up here should be improved. That is not good enough quality for readers of Nature Communications.

Reviewer #3:

Remarks to the Author:

This study used projected climate data and population data with age structure to estimate future population exposure to heat due to climate change. This is an interesting study but more work is needed before publication.

1. My foremost concern is the clarity of the methods and results.

Methods:

1) It took me some time to understand how the authors estimate the γ^* parameter, probably because they used different notations (e.g., g and $g(c)$).

2) What age structure did the authors use for further calculation? 5-year interval?

3) In equation (3), when the authors performed the calculation inside the sum, they assumed all age groups had the same population growth factors. I think this limitation should be explicitly mentioned.

4) For population heat exposures, the "acute exposure" is a confusing concept. The authors refer to "acute" as the average frequency and intensity of extremely hot days during the entire 20 years. How

do these two mean acute effect? I asked this because "acute" and "cumulative" effect both have policy implications.

5) Why did the authors use different types of Tmax for #HD (dry bulb Tmax) and T95 (regular Tmax)? And why was CDD calculated using Tmean rather than Tmax, which actually can be consistent with the other two metrics?

6) The expression of H is problematic. The authors described it as a fraction (% increase compared to 2020). But in equation 13 it stands for the gridded individual-level exposure (if I understand it right).

7) In equation 14, what does ϵ mean? Why can the fractional change in the population exposure be mathematically calculated as the sum of these three terms? What is the rationale?

Results:

1) Lines 125-126. How did the authors calculate the population-weighted exposures? It needs to be explicitly described in Methods.

2) Lines 134-135. Similarly, how did the authors calculate this? The Methods didn't mention anything about increase in the number of exposed population using a threshold.

3) Lines 132-134. No dashed line for C and D, and CDD is not an acute exposure based on the authors' definition.

4) Lines 179-182. What results (in what figure or table) were compared exactly? I'm not following. And the significance of difference should be noted in a table, rather than a simple notation in a sentence.

5) Figure 3. I find the x-axis is difficult to understand. Shouldn't it be continuous, if the authors expressed the changes with a smoothed curve? I mean, for example, for the upper right figure, what does the curve between "35-37.5" and ">37.5" mean? There is not any value between these two.

2. Lines 56-70. What do the authors want to express with these classifications? And they seem not to be mutually exclusive to me. For example, for (ii) studies assessing different dimensions and for (iii) studies with various spatial scales all most likely project future exposures. In addition, the fourth type of study may primarily involve exposure-response function investigation and projection. I guess I'm just getting confused where the authors want to go with this paragraph; at least it is not that relevant to population aging.

3. Lines 72-78. Difficult to read and understand.

4. Line 112. I think this is only true for Asia as I can tell from Figure 1. Southern Europe is in red color, meaning an aging population as well. For Africa I don't think it's true only based on the higher % of older population in northern Algeria.

5. Lines 120 and 125. Should be Figures 2A, 2C, and 2E, and Figure 2B, 2D, and 2F.

6. Line 213. Should be PDD.

7. There are no limitations in Discussion. At least there was a strong assumption about the same exposure level in the same grid with no adaptation (e.g. air conditioning).

8. Figure 5. The title of the lower-right one should be 70+.

9. Is there some sort of typo in equation 8?

Responses to the reviewers' comments

Thank you again for submitting your manuscript "Aging in a warming world: global projections of cumulative and acute heat exposure of older adults" to Nature Communications. We have now received reports from 3 reviewers and, on the basis of their comments, we have decided to invite a revision of your work for further consideration in our journal. Your revision should address all the points raised by our reviewers (see their reports below).

It is imperative you clarify the novel contribution of the study, and address concerns about validity raised by R1.

When resubmitting, you must provide a point-by-point response to the reviewers' comments. Please show all changes in the manuscript text file with track changes or colour highlighting. If you are unable to address specific reviewer requests or find any points invalid, please explain why in the point-by-point response.

Important: In addition to the above, you must comply with the following editorial requests; we will not be able to proceed with your revised manuscript otherwise. Please also see the Nature Communications formatting instructions, which you may find useful while preparing your revised manuscript.

We hope to receive your revised paper within three months, but we understand that revisions may take longer. Please let us know if you find that the revision process will take substantially more time.

Thank you very much for the opportunity to revise the manuscript. Please find here below point-by-point responses to the reviewers' comments.

Reviewer 1

Comment	Response
This paper combined the recent CMIP6 climate projections with downscaled population aging projection to estimate the future heat exposure for different age groups using three different metrics. It also provides a detailed decomposition of the drivers of old-age individuals' heat exposure, considering three co-occurring trends: future heat exposure, overall population growth, and demographic changes in the age structure. Overall, this paper addresses a much-needed topic on the amplifying role of population aging on heat exposure under climate change. I have the following comments for the authors to consider.	We thank the reviewers for their positive feedback and constructive comments. Please find here below point-by-point responses.

1. A major question is that despite the usage of three different metrics, the results look quite similar and all metrics convey quite the same message. It's unclear to the reviewer why presenting all three metrics throughout the main text? A suggestion is to keep one as the main analysis and put others in the supplementary (or just one main figure showing the similarity for results from different metrics).	The purpose of the analysis is to assess the change in both cumulative and acute older adult heat exposure. Throughout the paper, we refer to results based on the three different metrics to highlight how changes in exposure are heterogeneous across metric and region. For instance, it is important to note that while the exposure growth is consistent across indicators, cumulative exposure (PDD) indicators are much more sensitive to climate change than the intensity of acute exposure (PD95). These results highlight the fact that climate change will make average temperatures hotter for everyone – with the strongest repercussion for places that today have milder climates. Conversely, hotter regions will remain hotter also in the future – and thus in those areas the key drivers of the intensity of acute exposure growth will thus be demographic growth and change. The frequency of acute exposure stands in the middle, with some areas witnessing a surge in hot days due to climate change, but other areas where hot days are already more frequent witnessing a population aging and therefore increased elderly exposure.
2. Another question is regarding the choice of the Shared Socioeconomic Pathway (SSP) scenarios. This study uses only two Shared Socioeconomic Pathway (SSP) scenarios – SSP245 and SSP585. The SSP585 per se is not regarded as the 'business as usual' scenario any more in CMIP6 simulations, due to the existing climate policies. Given the availability of the SSP126 and SSP370 scenarios in the NEX-GDDP-CMIP6 datasets, it is unclear why the authors only chose two of them. The inclusion of more scenarios could have provided a more comprehensive understanding of the potential impacts of climate change on the aging population.	We have added SSP126 and SSP370 to the analysis. The former describes a scenario of limited climate change impacts (strong emission reductions) and sustained demographic growth, while the latter depicts a scenario of business as usual climate change (high impacts) and very strong demographic growth. The analysis entailed producing downscaled age-stratified population projections and extracting climate data and building heat-related indicators for these two additional scenarios. In the revised manuscript, all figures now include all four scenarios, resulting in a more comprehensive range of heat exposure projections. The four scenarios are briefly introduced in both the introductory and in the Methods section, including a description of the demographic change and climate change assumptions underlying each scenario. In general, we now refer to SSP245 as the "base" scenario (presenting it in Figure 1, 3, and 4, and showing results with the other three scenarios in the SI Appendix). Nonetheless, Figures 2 and Table 1 allow the reader to compare among scenarios. The text then comments upon the different scenarios and discusses the findings based on the underlying scenario assumptions. We believe that presenting the results across four distinct scenarios and demonstrating generally similar patterns throughout – while dense – provides a more convincing and rigorous assessment of our research question.
3. This may be the most critical concern. The conclusion of the amplifying role of population aging on the heat exposure under climate change at the global scale	Thank you for this very useful comment. We have compiled a much more extensive database of existing age-stratified sub-national population projections from different sources and for

is already covered in many previous studies, such as the Lancet Countdown Report on Health and Climate Change. Thus, the novelty of this study lies in the usage of the downscaled population aging projections under SSP scenarios in estimating future heat exposure, the validity of which is a major concern. Given the availability of age-stratified population projections is limited for some regions, validation of the imputation method used in this study needs further discussion. To understand the accuracy of this imputation method in the US and Europe in Figure 5 (i.e., understand the MAPE range), it is suggested to present the US and Europe specific population aging maps to double check the spatial pattern of population aging. The Striessnig et al. paper emphasized the importance of within country spatial variations of the population aging trends under different SSPs. If the 1-km downscaled population aging projection data is acceptable, it will have huge implications for many researches in the field and thus warrants further carefully examination.

different countries, including the US (two independent sources), EU nations, India, the UK, and China.

We then produced an array of additional comparison figures and metrics (see SI Appendix and Methods section), including Table 3 in the Methods which summarizes the estimated discrepancy – in both absolute and relative terms – in each country and in each scenario for each source.

Our main conclusion is that no projection study can claim to generate “correct” results and thus serve as a “gold standard” (after all, we are trying to simulate the future). However, by comparing projection studies focused on specific countries that have used the SSP assumptions with our grid cell-level projections, we observe a quite high concordance. For instance, comparing our estimates with Striessnig et al. (USA projections), percentage error rates in the total number of individuals aged 69+ always remain below 2%.

The U.S. is an emblematic case of how different modelling studies can lead to very different conclusions. In the U.S. case, we have access to two independent, reputable (peer-reviewed) county-level sources, and look at the “average local percentage error”) i.e. a metric expressing the accuracy in the allocation of population change to each county).

Comparing our projections with these two US projection studies, we calculate local-level discrepancies of around 22% w.r.t. Striessnig et al., while the same metric climbs to 60-80% w.r.t. the Hauer study, showing the relevance of the different sub-national projection and downscaling approaches adopted in the two studies (which in principle simulate the same SSP scenarios).

Our gridded, age-stratified projections also compare satisfactorily with two independent India and China-focused sub-national, SSP-consistent, age-stratified population projection studies. In particular, in China (Chen et al.) we find higher percentage error rates in the total number of individuals aged 69+ (13-22%), but only slightly larger local-level discrepancies (19-26%). This suggests modest? Considerable? uncertainty in absolute fertility and mortality rates, but far less uncertainty over the spatial distribution of older individuals in Chinese provinces. In India, for which only SSP2 is modelled by KC et al., we calculate a percentage error rate in the total number of individuals aged 69+ of 7% and a local-level discrepancy of 25%.

Additionally, in studies that relied on own assumptions (rather than the SSP framework) to produce age-stratified population projections (e.g. the European Union and the United Kingdom), we observe that the discrepancies are reduced when comparing them with our projections for SSP3. This leads to

	percentage error rates in the total number of individuals aged 69+ always remain of 2.5% and local-level discrepancies in the range of 28-37%. On the other hand, the other SSPs diverge more considerably, suggesting different underlying demographic modelling assumptions. In sum we conclude that our age-stratified sub-national population projections compare well with a set of independent, SSP and not SSP-compliant, sub-national projection studies, albeit some uncertainties persist, chiefly in the spatial distribution of elderly individuals. The newly added figures in the SI appendix are also reveal insights into spatial “accumulation” dynamics of the discrepancies. Yet, given the rather “discrete” changes in heat exposure over space (e.g. climate generally does not change abruptly between the border of two counties), we conclude that these uncertainties do not invalidate the findings of our study.
4. In calculating Cooling Degree Days (CDD), why using daily max temperature as the metric and why 37.5 degree C as the threshold? Other papers have used other metrics such as daily mean temperature above 18.3 degree C as the definition of CDD (e.g., this Nature Sustainability paper: https://www.nature.com/articles/s41893-019-0441-9)	The cooling degree metric we use – as discussed in our Methods section– is based on a 24°C Celsius, which is a standard value used in the literature. For instance, the European Commission uses this threshold value for calculating CDDs: https://ec.europa.eu/eurostat/statistics-explained/index.php?title=Heating_and_cooling_degree_days_-_statistics#Heating_and_cooling_degree_days_at_EU_level By contrast, the 37.5°C threshold is used for the ‘hot day’ metric definition, and it is not related to the CDD metric. The 37.5°C Celsius metric, as discussed in the paper, is based on a recent study in Lancet Planetary Health (https://www.thelancet.com/journals/lanplh/article/PIIS2542-5196(21)00079-6/fulltext) that identifies this threshold as a critical “upper temperature threshold of life”.

Reviewer 2

Comment	Response
This article examined future climate change risks in world scale. They considered climate, demographic and socioeconomic scenarios in projection of heat exposure. The results highlighted aging society in Africa and Asia are most vulnerable to future climate change. In my opinion, this article may include important contents that may be interesting to some readers, but this article is not novel at all and poorly explained about their results with other data. In addition, I do not think	We thank the reviewer for the critical feedback and the very useful comments. We have revised accordingly, and believe the manuscript is stronger as a result. Please find here below point-by-point responses.

authors know previous articles which already studied important factor such as demographic changes and adaptation using valuable statistic methods. Thus, I recommend to reject this paper.	
1. In introduction part, authors need to read and refer more previous projection studies. Based on my knowledge, there are more projection articles related health effects under RCP, SSP scenarios in important regional scale, such as North America, Europe area, Asia (around China) and etc. They already dealt with contents addressed in this articles. Authors need to find them, introduce some of them (most related articles to this) briefly and show difference between them and your work in the introduction part.	We now clarify that our study is not an examination of the health implications of heat exposure, nor it is a systematic literature review paper. Rather, we raise these two issues in the introduction as a way to motivate the importance of our study, and contextualize our focus on older adults. We are calculating exposure levels under different metrics and scenarios for a particularly vulnerable stratum of the population, older adults ages 69+. We hope that our results and the output exposure data (hosted at https://doi.org/10.5281/zenodo.7859233) will ultimately be used as an input in future epidemiological studies. However, that is not an explicit goal of our paper, as we cannot predict how often our data will be used by members of the larger research community However, to address concerns regarding the originality of our contribution, we more clearly state that our work is unifying two disparate but rarely overlapping research initiatives—changing climate, and population aging.
2. It is interesting that authors use CDD and HDs. Why do you use this term?	CDDs = “cooling degree days” HD = “hot days” Both acronyms are defined upon first mention, in the final paragraph of the introductory section. Thereafter, we refer by the metrics using the acronyms, for brevity’s sake.
3. In method section, is there any logical reason you choose 69+ as elderly population? Please explain.	We have both substantive and methodological reasons for our use of the 69+ cutpoint. First, while there is no standard definition of ‘elderly population’, many studies and national definitions have traditionally relied on the 65 years threshold. However, more recently, governmental reports concluded that “given changes in remaining life expectancy over time, age 70 can be thought of as the new age 65” (source). Therefore, we deem age 69 as relevant threshold to consider, and we now include this clarification in the introductory part of the methods section citing the report in question. Second, our data are structured into age groups that require our use of the 69+ cutpoint. The analysis starts from 5-year

	age structure from the age of 0 to the age of 80+ (thus 18 age groups) as derived from the WoldrPop age-stratified grid-cell level population count data that is used as the baseline for projections in the analysis. Then, these groups are aggregated into three broad groups generally representing youth, adulthood, and old age (0-19, 20-69, 69+), as presented in the paper.
4. In method section, authors mentioned that age-stratified population projection are available for US and Europe. Thus they use that data for rest of the world. But this is not correct. You can find future age structure data for each country in UN website. Please find more available data. As you already know, age structure in Asia is far different from Europe and USA. So, applying same strategy in analysis is unreasonable and not logical here.	We apologize if our original text was unclear or misleading. We now describe it more clearly. We now clarify that there are few countries for which future age structure data at the sub-national level (e.g. gridded, counties, or administrative units) are available. these sub-national data are crucial to COMPARE AND VALIDATE our spatial downscaling approach, which seeks to translate CURRENT age-stratified grid-cell level population counts (from WorldPop) into FUTURE age-stratified grid-cell level population counts based on the SSP demographic database, which provides FUTURE age-stratified COUNTRY-LEVEL population counts. To the best of our knowledge, there exists no global, sub-national, age-stratified dataset of demographic projections. The EU and USA data (which in the revised version of the manuscript are enriched with comparison data from additional countries for which we could retrieve sub-national projections) are used as a source of comparison with our sub-national estimates. These comparison data were retrieved for countries/regions where other sources of FUTURE age-stratified sub-national population counts have already been produced with the aim of evaluating the degree of consistency (irrespective of different projection approaches and embedded assumptions, such as on fertility and mortality rates, as well as migration). Refer to the Methods for a description and the SI Appendix for the validation/comparison results. .
5. As you presented in Figure 4, major factor here is aging. Previous studies mentioned about the importance of aging in climate change (please find “Projection of future temperature-related mortality due to climate and demographic changes” in EI journal). I do not think authors know about this.	Our goal is not to explore the impacts of climate change on mortality. Rather, to motivate the importance of our topic, we summarize data showing that the effects of heat on mortality are especially large for older adults. This begs for an exploration at the population level of the co-occurrence and spatial patterning of two social patterns: climate change and population aging. .
6. Except factors author mentioned here, other socioeconomic factors affect future health effect in climate change. You can find more data about that and can apply in	Thank you for this important observation. In our Discussion, we now underscore that the impacts of extreme heat are harmful not only to older adults, but to vulnerable

every country. This model you used here is too simple now.	populations with insufficient economic and structural resources. . We encourage future studies to explore the complex ways that heat affects areas with both large shares of older and lower-income persons. However, this goal is beyond the scope (and page constraints) of the current manuscript. We further recognize that such an assessment would require data that may not yet exist. To our knowledge there is no globally-relevant, age-stratified, mortality assessment/dose-response functions published that could be used to conduct such assessment. Studies that we are aware of are highly context-specific or are not age-group specific, e.g. https://www.thelancet.com/journals/lanplh/article/PIIS2542-5196(22)00139-5/fulltext , https://www.nature.com/articles/srep28161
7. Exposure is not always related to health effects. The degree of effects will be different based on each country's developments and adaptation strategies. Like mentioned in #6 comments, there are specific data (insurance, electricity usage...) which related to exposure-health correlation. Please address that or at least mention that in limitation.	Agreed, see response above. We mention AC use and electricity as well as other adaptation strategies in our Conclusion, citing studies on the topic where relevant.
8. Did you use R software for calculation? Add the sentence what software you used for your analysis if there is any.	Yes, the whole analysis is carried out in R. The source code is available on Github. We now include a short paragraph on the software/packages utilisation in the methods section of the paper.
9. Figure presentations: Figures you showed up here should be improved. That is not good enough quality for readers of Nature Communications.	We have improved the quality of the figures by using vectoral versions that ensure higher resolution, as well as making small graphical amendments. We will consult with the Nature Communications Editorial team should further adjustments be required.

Reviewer 3

Comment	Response
This study used projected climate data and population data with age structure to estimate future population exposure to heat due to climate change. This is an	We would like to thank the reviewer for the positive feedback and the very useful comments. Please find here below point-by-point responses.

interesting study but more work is needed before publication.	
1. My foremost concern is the clarity of the methods and results. Methods: 1) It took me some time to understand how the authors estimate the γ^* parameter, probably because they used different notations (e.g., g and $g(c)$).	We have added some clarifications to the section, e.g. that $g(c)$ identifies each of the set of grid cells that belong to country c.
2) What age structure did the authors use for further calculation? 5-year interval?	The analysis starts from 5-year age structure from the age of 0 to the age of 80+ (thus 18 age groups) as derived from the WorldPop age-stratified grid-cell level population count data that is used as the baseline for projections in the analysis. Then, these groups are aggregated into three broad age groups that represent youth, adulthood, and old age (0-19, 20-69, 69+), as presented in the paper.
3) In equation (3), when the authors performed the calculation inside the sum, they assumed all age groups had the same population growth factors. I think this limitation should be explicitly mentioned.	The intuition of the reviewer is partially correct, in so far that the gridded population projections from Gao et al. refer to the total grid cell population (i.e. they are not age stratified). Nevertheless, the age-stratified growth rates are derived from the country-level population projections from Lutz, which allow inferring grid cell-level age-stratified growth rates. The implied limitation of this downscaling procedure is already mentioned after Equation 5: "The limitation of this approach is its assumption of geographically homogeneous, within-country trends by age stratum."
4) For population heat exposures, the "acute exposure" is a confusing concept. The authors refer to "acute" as the average frequency and intensity of extremely hot days during the entire 20 years. How do these two mean acute effect? I asked this because "acute" and "cumulative" effect both have policy implications.	Cumulative exposure is measured by Cooling Degree Days (CDDs). To clarify, a synonym of cumulative could be "chronic", or the recurrent heat exposure accumulated/experience by an individual through the year (this is now also mentioned in the paper, see revised last paragraph of the introductory section). Cumulative/chronic exposure is therefore quantified based on average daily temperatures. On the other hand, the acute exposure refers to short-lived peak times of exposure, and therefore it is measured with metrics constructed with the daily maximum temperature. For acute exposure we differentiate between both the frequency of acute exposure (how often a given maximum temperature threshold is overpassed during the year, i.e. the number of hot days), and the intensity (i.e. how intense is this acute/maximum heat exposure), based on the 95th percentile of maximum temperature metric.
5) Why did the authors use different types of T_{max} for #HD (dry bulb T_{max}) and T_{95} (regular T_{max})? And why was CDD	The answer to comment (4) above partially addresses this comment. To elaborate, it is worth specifying that is standard

calculated using Tmean rather than Tmax, which actually can be consistent with the other two metrics?	practice to calculate cooling degree days (CDDs) based on mean temperature, rather than max (source). In addition, the Tmax used for #HD and T95 is the same, (.e. dry bulb temperature in both cases).
6) The expression of H is problematic. The authors described it as a fraction (% increase compared to 2020). But in equation 13 it stands for the gridded individual-level exposure (if I understand it right).	The reviewer is right in understanding H_g as the gridded individual-level exposure (as seen in Equations 13 and 14). We agree that instead Equation 12 seem to suggest that that H_g describes the total local exposure (the product of exposure, population, and age group fraction of the population). Therefore, we renamed H_g in Equation 12, to E_g, to remain coherent with the notation of Equations 13 and 14.
7) In equation 14, what does ϵ mean? Why can the fractional change in the population exposure be mathematically calculated as the sum of there three terms? What is the rationale?	The ϵ summarise the three terms that are expanded in the right hand side of the formula. In turn, the total exposure change ($\Delta E_r/E_r$) can be expressed as the sum of the three terms because each term represents the total exposure change (from the current total exposure) in response to the projected change in each individual component (climate metric, elderly population fraction, total population). For instance, ϵ^A, describes the change in total exposure owing only to the changes in elderly population fraction, hence representing one of the three (additive) drivers of total exposure change.
Results: 1) Lines 125-126. How did the authors calculated the population-weighted exposures? It needs to be explicitly described in Methods.	Population-weighted exposures are calculated by weighting climate indicators (e.g. CDDs or TMAX95) by the number of people in each grid-cell when computing statistics. This approach ensures that locations where higher population densities are found, like cities, have a higher relative weight than uninhabited locations(e.g. the desert or forests) in the final metrics computations. This is now described in greater detail in the Methods section.
2) Lines 134-135. Similarly, how did the authors calculated this? The Methods didn't mention anything about increase in the number of exposed population using a threshold.	The estimated populations exposure numbers reported in 134-135 summarise and are directly inferred from the underlying numbers in Figures 2A-2C-2E. We clarify this material in the manuscript.
3) Lines 132-134. No dashed line for C and D, and CDD is not an acute exposure based on the authors' definition.	Dashed lines have been added for those panels C and D. We also clarified in the paper/caption that we refer to both cumulative and acute exposure with these dashed lines, depending on the metric considered.

4) Lines 179-182. What results (in what figure or table) were compared exactly? I'm not following. And the significance of difference should be noted in a table, rather than a simple notation in a sentence.	We added a Table with all the details (Table SI1) to clarify this additional test. We now make explicit reference to the table in the paragraph discussing the t-tests carried out.
5) Figure 3. I find the x-axis is difficult to understand. Shouldn't it be continuous, if the authors expressed the changes with a smoothed curve? I mean, for example, for the upper right figure, what does the curve between "35-37.5" and ">37.5" mean? There is not any value between these two.	We agree. As we are using discrete exposure thresholds in the figure, we have slightly changed the visualisation into a discrete range of values graph. The results remain unchanged.
2. Lines 56-70. What do the authors want to express with this classifications? And they seem not to be mutual exclusive to me. For example, for (ii) studies assessing different dimensions and for (iii) studies with various spatial scales all most likely project future exposures. In addition, the fourth type of study may primarily involve exposure-response function investigation and projection. I guess I'm just getting confused where the authors want to go with this paragraph; at least it is not that relevant to population aging.	Thanks, we have revised the literature classification, also selecting studies that are more directly related to elderly heat exposure.
3. Lines 72-78. Difficult to read and understand.	Rewritten and fixed.
4. Line 112. I think this is only true for Asia as I can tell from Figure 1. Southern Europe is in red color, meaning an aging population as well. For Africa I don't think it's true only based on the higher % of older population in northern Algeria.	Figure description revised.
5. Lines 120 and 125. Should be Figures 2A, 2C, and 2E, and Figure 2B, 2D, and 2F.	Fixed.
6. Line 213. Should be PDD.	Fixed.
7. There is no limitations in Discussion. At least there was a strong assumption about the same exposure level in the same grid with no adaptation (e.g. air conditioning).	We have rewritten the Discussion section, and added limitations paragraph which acknowledges that our exposure estimates are "baseline" exposure, before adaptation occurs. We also highlight some opportunities to combine our estimates with adaptation data to obtain "net exposure" estimates.
8. Figure 5. The title of the lower-right one should be 70+.	Fixed.
9. Is there some sort of typo in equation 8?	Yes, the "1" was a typo, thanks for spotting it.

Reviewers' Comments:

Reviewer #1:

Remarks to the Author:

The authors have done a great job in addressing my previous comments. Specifically, focusing on the projection of heat exposure rather than the health effects at full global coverage would be an important contribution to the field. The validation of aging projections using the best available evidence from a few countries also strengthens this study. I thus recommend the publication of this study.

Reviewer #2:

Remarks to the Author:

Based on your responses to the reviewers' comments, I think it is good enough to publish in Nature Communications. Thank you for your efforts.

Reviewer #3:

Remarks to the Author:

I'm still having problems understanding the interconnection between formula 12, 13, and 14. I think the authors need to write this part in more details for readers to understand.

1. What does equation 12 do in the following calculation?
2. It is still "H" above equation 12.
3. If the authors changes H to E, then what does that meteorological heat index H mean? This is the only place throughout the paper with this term. And how was it calculated? Why can it be applied in the calculation of ϵ_r ?
4. For equation 14, the authors did answer my question of "what does ϵ_r mean". It needs to be explicitly stated.
5. Also for equation 14, I don't understand the meaning of "gridded sum of weighted fractional change" for each of the three terms. I mean, what does it mean by summing all gridded fractional changes? Why would it generate the overall fractional change? It doesn't make too much sense to me.
6. Also in equation 13, 14, and the line below equation 14, the "Hg" needs to be defined.

Responses to the reviewers' comments

Reviewer 1

Comment	Response
The authors have done a great job in addressing my previous comments. Specifically, focusing on the projection of heat exposure rather than the health effects at full global coverage would be an important contribution to the field. The validation of aging projections using the best available evidence from a few countries also strengthens this study. I thus recommend the publication of this study.	Thank you very much for the positive feedback and – once again – for the constructive and useful comments from the previous review round.

Reviewer 2

Comment	Response
Based on your responses to the reviewers' comments, I think it is good enough to publish in Nature Communications. Thank you for your efforts.	Thank you very much for the positive feedback and – once again – for the constructive and useful comments from the previous review round.

Reviewer 3

Comment	Response
I'm still having problems understanding the interconnection between formula 12, 13, and 14. I think the authors need to write this part in more details for readers to understand.	Thank you very much for pointing out these issues. We have sought to extensively rewrite the section of the Methods with the aim of more clearly presenting and describing the mathematical formulation of the heat exposure quantification, projection, and drivers decomposition, while also partially revising the population downscaling and projection equations to ensure consistency in the notation (pages 15-18 of the revised manuscript). We invite you to read through the revised description to check if ambiguities and doubts are now solved. Please note that the equation numbers have changed in the revised version of the manuscript due to the use of sub-equations (e.g. Eq 7a, 7b, 7c, etc.) and re-structuring of the

	paragraphs. Specifically, former Equation 12 (now Equation 10) was moved to section “Decomposing exposure change into the contributions of its drivers” and reformulated with coherent notation and subscript to link it to former Equations 13-14 (now equations 8 and 11, also revised).
1. What does equation 12 do in the following calculation?	As detailed in the revised manuscript, Equation 12 (now Equation 10) describes the grid cell level fractional changes in heat exposures between the current and potential future climates. The methods then explain how these changes can be fractionally decomposed (Equation 11) into the joint determinants of change in regional-scale exposure, i.e. the weighted averages of grid-cell level fractional shifts in age structure, population size, and meteorology (climate).
2. It is still "H" above equation 12.	Fixed by changing H to E. Apologies, this was a typo coming from the previous revision. To solve residual ambiguities, H and E are now explicitly defined (paragraph below equation 7c).
3. If the authors changes H to E, then what does that meteorological heat index H mean? This is the only place throughout the paper with this term. And how was it calculated?	As mentioned above, H and E are now explicitly defined (paragraph below equation 7c). Specifically, “let H denote climate change-driven increases in heat in Equation 6 , and E denote the population exposures in Equation 7.” Calculation details are reported in those corresponding equation.
4. Why can it be applied in the calculation of E_r ? For equation 14, the authors did answer my question of "what does ϵ_r mean". It needs to be explicitly stated.	As now explicitly explained in the text before Equation 8, “Generalizing the foregoing results to the scale of regions, r , population exposure of older individuals in any current or future climate scenario, $E_{r,s}$, is simply the sum over grid cells of Equation 7”. I.e., the regional aggregation of grid-cell level exposures.
5. Also for equation 14, I don't understand the meaning of "gridded sum of weighted fractional change" for each of the three terms. I mean, what does it mean by summing all gridded fractional changes? Why would it generate the overall fractional change? It doesn't make too much sense to me.	The sentence was rephrased to clarify this ambiguity (text around Equation 11): “Algebraically decomposing the fractional change in regional-scale exposure yields the sum of three terms that capture the weighted averages of grid-cell level fractional shifts in age structure, population size, and meteorology [...] in which the weights are grid cells’ fractional contributions to total exposure”.
6. Also in equation 13, 14, and the line below equation 14, the "H _g " needs to be defined.	Here, H_g defines the value of each heat exposure metric (H) at each grid cell g (Equation 8 and paragraph above it).

Reviewers' Comments:

Reviewer #3:

Remarks to the Author:

The authors have successfully addressed my concerns. Therefore, I think this manuscript is ready for publication.